# A DNA Prime Immuno-Potentiates a Modified Live Vaccine against the Porcine Reproductive and Respiratory Syndrome Virus but Does Not Improve Heterologous Protection

**DOI:** 10.3390/v11060576

**Published:** 2019-06-25

**Authors:** Cindy Bernelin-Cottet, Céline Urien, Maxence Fretaud, Christelle Langevin, Ivan Trus, Luc Jouneau, Fany Blanc, Jean-Jacques Leplat, Céline Barc, Olivier Boulesteix, Mickaël Riou, Marilyn Dysart, Sophie Mahé, Elisabeth Studsrub, Hans Nauwynck, Nicolas Bertho, Olivier Bourry, Isabelle Schwartz-Cornil

**Affiliations:** 1VIM, INRA, Université Paris-Saclay, Domaine de Vilvert, 78350 Jouy-en-Josas, France; bernelin.cottet.cindy@hotmail.fr (C.B.-C.); celine.urien@inra.fr (C.U.); maxence.fretaud@inra.fr (M.F.); christelle.langevin@inra.fr (C.L.); luc.jouneau@inra.fr (L.J.); nicolas.bertho@inra.fr (N.B.); 2VIM, EMERG’IN-Plateforme d’Infectiologie Expérimentale IERP, INRA, Domaine de Vilvert, 78352 Jouy-en-Josas, France; 3Laboratory of Virology, Faculty of Veterinary Medicine, Ghent University, Salisburylaan 133, B-9820 Merelbeke, Belgium; ivan.trus@gmail.com (I.T.); hans.nauwynck@ugent.be (H.N.); 4GABI, INRA-AgroParisTech, Université Paris-Saclay, Domaine de Vilvert, 78350 Jouy-en-Josas, France; Fany.blanc@inra.fr (F.B.); jeanjacques.leplat@inra.fr (J.-J.L.); 5Plate-Forme d’Infectiologie Expérimentale-PFIE-UE1277, Centre Val de Loire, INRA, 37380 Nouzilly, France; celine.barc@inra.fr (C.B.); olivier.boulesteix@inra.fr (O.B.); mickael.riou@inra.fr (M.R.); 6Pharmajet, 400 Corporate Circle Suite N, Golden, CO 80401, USA; marilyn.dysart@pharmajet.com; 7Unité Virologie et Immunologie Porcines, Laboratoire de Ploufragan-Plouzané-Niort, Anses, BP 53, 22440 Ploufragan, France; sophie.mahe@anses.fr (S.M.); olivier.bourry@anses.fr (O.B.); 8Vaccibody AS, Gaustadalleen 21, 0349 Oslo, Norway; estubsrud@vaccibody.com

**Keywords:** PRRSV, DNA vaccine, modified-live vaccine, antigen-presenting cell targeting, pigs

## Abstract

The porcine reproductive and respiratory syndrome virus (PRRSV), an RNA virus inducing abortion in sows and respiratory disease in young pigs, is a leading infectious cause of economic losses in the swine industry. Modified live vaccines (MLVs) help in controlling the disease, but their efficacy is often compromised by the high genetic diversity of circulating viruses, leading to vaccine escape variants in the field. In this study, we hypothesized that a DNA prime with naked plasmids encoding PRRSV antigens containing conserved T-cell epitopes may improve the protection of MLV against a heterologous challenge. Plasmids were delivered with surface electroporation or needle-free jet injection and European strain-derived PRRSV antigens were targeted or not to the dendritic cell receptor XCR1. Compared to MLV-alone, the DNA-MLV prime- boost regimen slightly improved the IFNγ T-cell response, and substantially increased the antibody response against envelope motives and the nucleoprotein N. The XCR1-targeting of N significantly improved the anti-N specific antibody response. Despite this immuno-potentiation, the DNA-MLV regimen did not further decrease the serum viral load or the nasal viral shedding of the challenge strain over MLV-alone. Finally, the heterologous protection, achieved in absence of detectable effective neutralizing antibodies, was not correlated to the measured antibody or to the IFNγ T-cell response. Therefore, immune correlates of protection remain to be identified and represent an important gap of knowledge in PRRSV vaccinology. This study importantly shows that a naked DNA prime immuno-potentiates an MLV, more on the B than on the IFNγ T-cell response side, and has to be further improved to reach cross-protection.

## 1. Introduction

The Porcine Reproductive and Respiratory syndrome virus (PRRSV), a positive single stranded RNA enveloped virus of the *Arteriviridae* family, is responsible for high economical losses in the swine industry. The PRRSV induces reproductive failures during late gestation in sows and respiratory disorders in neonates and in growing-finishing piglets, resulting in poor growth performance [1,2]. The PRRSV RNA genome includes 10 open reading frames (ORFs), which encode for seven structural proteins and 14 nonstructural proteins. Two distinct genotypes, now considered as separate species [3], exist which present only 60% genomic sequence identity, with PRRSV-1 being dominant in Europe and PRRSV-2 in America. Both species co-exist in Asia with frequent emergence of highly pathogenic strains. Among European strains, Eastern strains have emerged that display high virulence and cause large outbreaks, whereas circulating Western strains generally display low virulence and more insidious disease. The Eastern and Western strains are genetically divergent and grouped in subtypes 3 and 1 respectively [4]. This genetic diversity is related to the high dynamics of PRRSV genome, which continuously and rapidly evolves, generating new variants and expanding its diversity which is the main hurdle to effective prevention and control of PRRS through vaccination [4]. Modified live vaccines (MLVs) obtained by serial in vitro passages for attenuation are currently the most used vaccines, as they can reduce disease severity as well as the duration of viremia [5,6]. However, MLV efficacy is greater against homologous strains and declines dramatically for genetically distant heterologous PRRSV strains [5,6].

T cell-mediated immunity has been proposed to be involved in the heterologous protective efficacy of MLVs against diverging strains [7]. Interestingly, T-cell epitopes from different PRRSV ORFs have been described that are conserved through European distant strains [8]. In a parallel study, in order to enhance and broaden the T-cell mediated immunity induced by MLVs, we used a DNA-MLV prime-boost strategy with plasmids encoding PRRSV antigens (PRRSV-AG) including conserved T-cell epitopes (NSP1β, RdRp, M-derived antigens) as well as the B cell immuno-dominant nucleoprotein N from a recent Western European strain [9]. We found that the DNA prime broadened the T-cell response and potently enhanced the anti-N IgG response induced by MLV. Furthermore, when PRRSV-AGs were expressed in vaccibody (VB) platforms targeted to XCR1, a receptor selectively expressed by a dendritic cell (DC) subset across species [10], the anti-N IgG response, but not the IFNγ T-cell response, was further enhanced. In that first study, the plasmids were combined with cationic poly-lactoglycolide acid (PLGA) nanoparticles (NPs) and administered intra-dermally with surface electroporation (EP). NPs may affect the efficacy of the DNA-MLV prime-boost strategy and the DC-targeting outcome, and they add complexity and cost to the manufacture step. In addition, surface EP, although very efficient, has currently not yet been adapted to vaccine delivery in veterinary field conditions. A painless intradermal jet delivery device has been shown to be particularly efficient to induce high neutralizing antibody responses in pigs using DNA encoding the influenza hemagglutinin antigen [11] and such a convenient delivery, now licensed for human use, could be adapted to field use for veterinary applications. Therefore, in the present study, we evaluated the T and B cell responses induced by a DNA-MLV prime-boost strategy with naked DNA vaccines administered using surface EP compared to jet delivery and coding for PRRSV-AGs targeted or not to XCR1. We also assessed whether such a DNA-MLV prime-boost strategy would enhance the cross-protection against a distant European PRRSV strain. 

## 2. Material and Methods

### 2.1. Antibodies (Abs)

The anti-pig IFNγ mouse P2G10 mAb (capture), biotinylated anti-pig IFNγ P2G11 mAb and alkaline phosphatase-conjugated streptavidin were from MabTech AB (Nacka Strand, Sweden). The anti-mCherry rabbit IgG was purchased from Rockland, Tebu-bio SAS, Le Perray en Yvelines, France). The Alexa Fluor 594-conjugated goat anti-rabbit IgG was from ThermoFisher Scientific, Waltham, MA, USA.

### 2.2. Viruses

The PRRSV-1, subtype 1, 13V091 (FL13), and 07V063 (FL07) strains have been described previously [12]. FL13 was attenuated by 73 in vitro passages in the MARC-145 cell line, clone F4, kindly provided by Prof. Dr. M. Pensaert (UGent, Belgium). This attenuated strain, designated here as MLV-FL13b, was unable to infect primary alveolar macrophages, even after in vivo passage (unpublished). The FL07 strain was propagated on specific-pathogen-free alveolar macrophages.

### 2.3. DNA Vectors 

An mCherry encoding plasmid (mCherry2-N1, here referred to as pmCherry) was obtained from Addgene (Cambridge, MA, USA) under MTA. A firefly luciferase expression plasmid (pLuc) was kindly provided by Stéphane Biacchesi (INRA, Jouy-en-Josas, France). The plasmid vectors encoding untargeted PRRSV antigens (AG) are based on a pcDNA3.1 vector: pN, pNSP1β, and pRdRp vectors encoded for the native N, NSP1β, and RdRp viral proteins respectively and pGP4GP5M encodes a chimera that was derived from a previous publication [13] and transposed to the FL13 strain [14]. The construction of the vaccibody (VB) vaccine format has been previously described [15]; it includes a targeting unit, a dimerization unit made of a human Ig hinge and CH3 sequences of human IgG3, and an antigen unit. A VB with porcine XCL1 as the targeting unit was previously published by our group [16] and this same targeting unit sequence including the natural XCL1 leader sequence was used here. All sequences were codon-optimized according to the porcine species, synthesized by GenScript (Piscataway, NJ, USA) and cloned in a pUMVC4a vector (Aldevron, Fargo, ND, USA). The four VB plasmids encoding PRRSV AG are pXCL1-N, pXCL1-GP4GP5M, pXCL1-NSP1β1 (AA1 to 202 of NSP1β), and pXCL1-NSP1β2 (AA182 to 383 of NSP1β) and are further described in submitted manuscript 1 [9]. The NSP1β sequence had to be separated into two parts to allow expression of the chimera. Vaccibody constructs encoding for RdRp did not lead to detectable expression upon transfection in HEK293 cells and therefore were not used [9]. Plasmid productions for immunization were prepared using endotoxin-free NucleoBond^®^ EF kit (Nagel Macherey GmbH, Düren, Germany) according to the manufacturer’s instructions and were stored at −20 °C until use.

### 2.4. Pig Studies, Ethic and Authorizations

All experiments were conducted in accordance with the EU guidelines and the French regulations (DIRECTIVE 2010/63/EU, 2010; Code rural, 2018; Décret #2013-118, 2013). All experimental procedures were evaluated and approved by the Ministry of Higher Education and Research. The experiments with the pLuc plasmid (skin cell transduction) were approved by the COMETHEA ethic committee under the APAFIS notification number 2017120408328722 v2 (23 February 2018) in accordance with national guidelines on animal use and performed at the Animal Genetics and Integrative Biology unit (GABI-INRA), France, under the accreditation number for animal experimentation C78-719. Large White pigs (10 weeks of age) were obtained from UEPR-INRA-Rennes-Saint Gilles. The immunization experiments with PRRSV antigen-encoding plasmids was approved by the Comité d’Éthique en Expérimentation Animale Val de Loire (CEEA VdL, committee #19) under the APAFIS notification number 2015051418327338 v10 (3 July 2017) and were conducted at the Plateforme d’infection Expérimentale PFIE-INRA, Nouzilly, France, (https://doi.org/10.15454/1.5535888072272498e12) under the accreditation number for animal experimentation D37-1753 and under A-BSL3 containment. Social material enrichment (balls, chains) was provided to maintain piglet welfare. Large White pigs (1 month of age) were obtained from the INRA conventional breeding unit Unité Expérimentale de Physiologie Animale de l’Orfrasière PAO-INRA, Nouzilly, France, working under the A37-1754 accreditation number for animal experimentation. The piglet mothers were checked to be PRRSV-seronegative with a commercial ELISA.

### 2.5. In Vivo Gene Transfer 

Pigs were anesthetized (2% isofluorane) to avoid animal discomfort with electroporation (EP) and to optimally control the quality of vaccine administration. Pig skin was cleaned with Vetedine^®^ soap (Vetoquinol, Magny Vernois, France), washed and dried. Part of the same experiment was submitted to publication for a study comparing plain pLuc DNA administration to DNA on cationic poly-lactoglycolide nanoparticles and patch administration [14]. Therefore, the values of the luminescence and of IL1β and IL-8 secretion induced by EP delivery are the same ones as in that paper. For EP delivery, intradermal injection of pLuc and pmCherry (100 µg in 100 µL saline) was done in the inguinal zone and EP was performed on the site of injection using the CUY 21 EDIT system (NEPA GENE Co. Ltd., Chiba, Japan). Disk electrodes (10 mm) were loaded with conductive gel (Alcyon, Paris, France) and 6 electric pulses were applied during 10 ms with 90 ms interval. Trolamine 0.6% (Biafine, Johnson & Johnson Santé Beauté, Issy-les-Moulineaux, France) was spread on the transfected zone right after administration. For needle-free jet delivery intradermal application (Tropis ID, Pharmajet, https://pharmajet.com/tropis-id/, Golden, CO, USA), pLuc and pmCherry (100 µg in 100 µL saline) were administered in the lower thoracic zone. This method of administration is designated as PJ in this study. After 24 h, pigs were euthanized and skin biopsies were collected using an 8-mm disposable biopsy punch (Kai medical, Tokyo, Japan) after cleaning of the skin with Vetedine soap and Vetedine 10% solution (Vetoquinol, Magny-Vernois, France). Control skin was harvested from the same zone (inguinal for the EP and thoracic zone PJ). 

### 2.6. Detection of pLuc Expression

Two 8 mm-diameter biopsies transfected with pLuc per pig and per mode of delivery were cut into small pieces and lysed with 150 µL of lysis reagent (Luciferase Assay System E1500, Promega, Madison, WI, USA). Bioluminescence was measured per each biopsy using the In Vivo Imaging System (IVIS-200, Xenogen, Caliper LifeSciences, Hopkinton, MA, USA) after adding 100 µl luciferin substrate. A region of interest (ROI) was manually selected and the intensity of luminescence (photons/sec/cm^2^) was recorded.

### 2.7. Detection of pmCherry Expression 

One biopsy from skin transfected with pmCherry was collected per pig (3 pigs) and per mode of delivery (EP and PJ) and fixed in buffered formalin 4%. Immunohistochemical staining and clearing were performed following the iDISCO+ protocol [17]. Biopsies were incubated 7 days in each antibody solutions. The primary antibody (anti-mCherry rabbit IgG) and the secondary antibody (Alexa Fluor 594-conjugated goat anti-rabbit) were diluted 1:500 in PBS. Before clearing, samples were counterstained with 4′,6-diamidino-2-phenylindole (DAPI, Sigma-Aldrich, Merk KGaA, Darmstadt, Germany) diluted at 5 µg/mL before clearing. Biopsies were mounted under coverslips (ThermoFisher Scientific, Waltham, MA, USA) using spacers made with Picodent twinsil speed 22 dental paste (Picodent, Wipperfürth, Germany) and sealed with Dentalon plus dental paste (Phymep, Paris, France). Images were acquired with a Leica SP8 two-photon microscope using a HCX IRAPO L 25X/0,95NA water immersion objective (Leica microsystems, Wetzlar, Germany). DAPI and Alexa Fluor 594 were both excited at 800 nm with a Chameleon Vision II laser (Coherent, Lisse, France) and fluorescence was detected by NDD HyD detectors (Leica microsystems, Wetzlar, Germany) with BP 525/50 and BP585/40 filters respectively. Image analyses were performed using the Fiji software (ImageJ2, Madison, WI, USA).

### 2.8. Detection of Inflammatory Cytokines in Pig Transfected Skin 

For cytokine production assessment, two biopsies (8-mm diameter each) per pig and per mode of delivery were carefully washed in PBS and each placed in 1 mL RPMI + 10% FCS + 1% antibiotic/antimycotic 100X cocktail (Gibco, ThermoFisher Scientific, Waltham, MA, USA) for 24 h. Concentrations of secreted cytokines were assessed by cytometric beads assay for simultaneous detection of 12 swine cytokines (IL-4, IFNα, TNFα, IL-2, IL-8, IL-13, IL-12, IL-6, IFNγ, IL-10, IL-17, IL-1β) as described [18]. 

### 2.9. Immunization of Pigs and Infectious Challenge

Pigs (1-month-old) were split into 8 groups including 8 or 9 piglets per group, respecting a balance between males and females, and taking their parental origin so that genetic biases between groups were minimal. The groups’ description and the immunization schedule are shown in Table 1. On day 0 (D0), 4 groups of pigs were anesthetized (2% isofluorane) and were injected with the DNA plasmids (400 µg each), either with EP or with PJ, as described above. In the case of EP, DNA in 400 µL was injected in 3 spots and in the case of PJ, DNA in 400 µL was injected in the costal zone in 4 spots. Trolamine 0.6% (Biafine^®^, Johnson & Johnson Santé Beauté, Issy-les-Moulineaux, France) was spread on the EP zone administration. On D34, the 4 DNA-vaccinated groups and an additional MLV-only group received MLV-FL13b (10^5.5^ TCID per pig) intramuscularly in 2 mL PBS + Ca^2+^Mg^2+^. On D63, all pigs including a control group were anesthetized with xylazine (Rompun^®^, Bayer SAS, Lyon, France) at 2 mg/kg and ketamine (Imalgene^®^, Merial, Boehringer Ingelheim, Ingelheim am Rhein, Germany) at 10 mg/kg by the intramuscular route and they received 2 × 10^5^ TCID_50_ FL07 per pig by the intra-nasal route (1 mL per nostril).

Appropriate equipment was used for the virus administration (protective cover-all clothing, gloves, and safety shoes). Body temperature was monitored by sensor chips placed under superficial skin muscle. Pigs were examined daily and monitored for dyspnea, coughing, nasal discharge, conjunctivitis, ear color, diarrhea, activity. They were euthanized on D88 by an overdose of Dolethal^®^ (Vetoquinol, Magny Vernois, France, 50 mg/kg).

### 2.10. Nasal Swab and Serum Collections

Nasal swabs were collected for viral detection in 500 μL RPMI and for antibody detection in PBS + cOmplete Protease Inhibitor Cocktail (Roche, Sigma-Aldrich, Merck, KGaA, Darmstadt, Germany), and they were immediately frozen. For virus detection, nasal swabs were collected D58, 69, 71, 73, 76, and 83. Sera were collected for antibody detection on D0, 34, 48, 58, 83, 88, and for viral detection on D58, 69, 71, 73, 76, 83.

### 2.11. Overlapping Peptides

Overlapping peptides (20 mers, offset 8) covering the NSP1β, RdRp, N and the GP4GP5M PRRSV antigens were synthetized by Mimotopes (Mimotopes Pty Ltd., Victoria, Australia, http://www.mimotopes.com). Upon receipt, the peptides were diluted in H_2_0:acetonitrile (50:50 vol) at a 5 mg/mL concentration and grouped as pools of peptides not exceeding 50 peptides: pool N (15 peptides), pool GP (GP4GP5M chimera, 40 peptides), pool NSP1β1 (47 peptides) and 2 pools RdRp1 (peptide 1 to 40), pool RdRp2 (peptide 41 to 80). A 20-mer peptide from the HIV polymerase was used as control. The list of the peptides is provided in Appendix A.

### 2.12. IFNγ Detection by ELISPOTS 

PBMCs were collected on D49 on three 8 ml Vacutainer^®^ CPT™ (BD-Biosciences, San Jose, CA, USA) by centrifugation at 1800× *g* for 35 min at 25 °C. PBMCs were washed with PBS + 1.3 mM citrate and re-suspended in X-vivo medium (Ozyme, Saint-Cyr-l’Ecole, France) + 50% FCS + 1.3 mM citrate and rested overnight. PBMCs were re-suspended in X-VIVO-20 medium supplemented with 2% FCS, 100 U/mL penicillin and 1 µg/mL streptomycin (culture medium) and counted for live cells. IFNγ-secreting T cells were detected using polyvinylidene difluoride (PVDF) membrane-bottomed 96-well plates (MultiScreen^®^_HTS_, Millipore, Merck KGaA, Darmstadt, Germany) coated with 15 µg/mL anti-porcine IFNγ (capture mAb) in PBS. PBMCs (2 × 10^5^) were plated per well and were stimulated with the different pools of overlapping peptides described above at a 10 µg/mL final concentration for 18 h, in triplicates. An HIV polymerase-derived peptide and ConA at 25 µg/mL were used as controls. After 18 h, the IFNγ-secreting cells were revealed by sequential incubations with 0.5 µg/mL biotinylated anti-IFNγ followed by 0.5 µg/mL alkaline phosphatase conjugated-streptavidin (MabTech AB, Nacka Strand, Sweden) and 1-Step^TM^ BCIP/NBT reagent (Sigma, Merck KGaA, Darmstadt, Germany). The spots were enumerated using the iSPOT reader from AID Autoimmun Diagnostica GmbH (Straßberg, Germany). Positive wells were considered if the mean spot numbers in the stimulated conditions were significantly superior to the spot numbers in the control peptide conditions (*p* < 0.05, paired *t*-test) and if >55 spots. The mean number of spots from stimulated minus control peptide wells was calculated.

### 2.13. Detection of Anti-N IgG and Anti-Envelope IgG

For the anti-N IgG detection, the sera prior challenge was assayed using the Ingezim PRRS 2.0 kit (Ingenasa, Eurofins-technologies, Bruxelles, Belgium) at a 1:40 dilution. The S/P ratios were calculated as follows: [OD sample minus OD negative control]: [OD positive control minus OD negative control], using the mean of the experimental negative controls (non-vaccinated group including 8 pigs, at each considered time point). For the detection of anti-envelope IgG (ELISA-Env), 96-well-plates were coated with PRRSV-1 envelope proteins. Individual pig sera (D58 and D88) were diluted 1:20 and incubated on coated plates at RT for 1 h. The plates were subsequently incubated with peroxidase-labeled anti-pig IgG antibody at 20 °C for 1 h. The peroxidase enzymatic activity was finally revealed using tetramethylbenzidine (TMB) substrate (ThermoFischer Scientific). OD was measured at 450 nm using a TECAN microplate reader Tecan Group Ltd., Männedorf, Switzerland. S/P ratios were calculated as for anti-N Abs.

### 2.14. Detection of Neutralizing Antibodies (NAbs) 

NAbs were quantified in sera on MARC-145 cells against the FL13b vaccine strain according to the Virus Neutralization Titer (VNT) method described in a previous paper [19]. 

### 2.15. Viral Detection by Specific qRT-PCR

For quantification of viral RNA in sera and nasal swab, viral RNA was first extracted from 100 µL biological fluid using NucleoSpin^®^ RNA Virus kit from Nagel-Macherey GmbH, Düren, Germany. A one-step qRT-PCR was performed using the iTaqTM universal probes One-Step kit (BIO-RAD, Hercules, CA, USA). Primers were designed to amplify selectively FL07 and not FL13 sequences, in the ORF1 gene: forward primer 5′-TGGCACAGAATCCGACAACA-3′, reverse primer 5′- GTCTAAGGCCTGCGCATCA-3′. A fluorescent probe was designed that hybridizes to the specific FL07 PCR product: FAM 5′-AGCTCGCCTCTGACTT-3′ TAMRA. A standard curve was made with four 10-fold dilutions of the FL07 inoculum (10^1^ to 10^4^ TCID_50_/mL), each spiked in control pig serum and extracted with the NucleoSpin^®^ RNA Virus kit. The qRT-PCR was performed with 2 µL of sample elution in 10 µL final mix and the cycling involved the following steps: reverse transcription at 50 °C for 10 min, denaturation at 95 °C for 3 min, amplification 40 cycles at 95 °C for 15 s, and 60 °C for 30 s. TaqMan run of experimental samples contained 2 replicates, mock pig serum RNA and H20. The reactions were carried out in a CFX Connect^TM^ light cycler (BIO-RAD). The number of TCID_50_-equivalent (TCID_50_ eq)/mL in each sample was determined using the FL07 RNA standard calibration curve. For each animal, the area under the curve (AUC) of the TCID_50_eq per ml over 25 days (viral RNA AUC) was calculated. The specificity of the qPCR reactions for FL07 and FL13 detection was checked on the respective standard curves and strictly no heterologous virus could be detected.

### 2.16. Statistical and Correlation Analyses

Data were analyzed with the GraphPad Prism 7.0 software (San Diego, CA, USA). The unpaired non-parametric two-way Mann–Whitney test was used to compare the inflammatory cytokine results, the luciferase, the ELISPOT, the Ab results and the viral AUC between 2 groups. The principal component analysis was done with 16 factors and included the individual viral RNA copies/mL at D69, 73 and 83 as well as the viral RNA AUC, the individual body temperatures at D65, the T-cell responses at D49 to the N, GP4GP5M, NSP1β, RdRp1 and RdRp2 pools, the total number of spots per pig (with and without the NSP1β responses), the number of T-cell antigens recognized per pig, the anti-N Ab at D58 and 83, and the anti-envelope Ab at D58 and D88. PCA graphics was produced using the FactoMineR R package (http://factominer.free.fr/). The correlation analysis between the different immune and viral parameters was done with a bilateral Spearman non-parametric test.

## 3. Results

### 3.1. Transduction Efficacy of Pig Skin Cells upon DNA Delivery with a Needle-Free High Pressure Jet Injection (Pharmajet Tropis-ID device, PJ) Versus Surface EP

In order to deliver our DNA vaccines in a convenient, well-tolerated and efficient manner, we compared the transduction efficacy of pig skin cells with a pLuc plasmid administered as plain DNA via needle-free high pressure jet injection (PJ) versus surface EP. Skin biopsies were harvested 24 h after pLuc delivery and processed for detection of luciferase expression by bioluminescence measurement. Figure 1A shows that PJ delivery induced luciferase expression reaching higher values (2.69 × 10^9^ ± 1.57 × 10^9^ p/s/cm^2^/sr) than surface EP delivery (1.30 × 10^8^ ± 1.46 × 10^7^, *p* < 0.005). 

In addition, we evaluated the in situ expression of a pmCherry plasmid, 24 h post-delivery with PJ and EP, using a whole tissue clearing strategy and detection with 2-photon microscopy. Figure 2 shows distinct mCherry-positive cells: keratinocytes (A, B), dermal cells resembling fibroblasts, cells with branched projections suggestive of DCs (C, D), and cells resembling adipocytes (E, F). No clear difference in the micro-anatomy of mCherry expression could be seen between EP and PJ delivery from three skin samples per injection mode. 

Skin biopsies were placed in flotation in culture medium for 24 h. The cytokine content was analyzed in a multiplex assay for detection of IL-4, IFNα, TNFα, IL-2, IL-8, IL-13, IL-12, IL-6, IFNγ, IL-10, IL-17, IL-1β. The DNA administration with PJ induced variable and low levels of IL-1β and IL- 8 (no statistical significance versus control skin of the same site) whereas EP induced significant amounts of both cytokines (Figure 1B). The expression of IL-2, IFNγ, and IL-13 was not or inconsistently detected and the expression of the other cytokines was not modulated upon DNA administration.

Altogether, PJ appears more potent than EP to transduce skin cells and induces lower inflammatory responses than EP.

### 3.2. IFNγ T-Cell Responses Induced by a Prime-Boost DNA-MLV-FL13b Strategy with DNA Encoding PRRSV-AGs Targeted or Not to XCR1 and Delivered with EP or PJ

We showed in ref. [9] that an improvement of the T-cell response breadth was achieved by a DNA-MLV prime-boost strategy, versus DNA-only or MLV-only. We selected DNA plasmids from the ones that were used in that study (see Material and Method section). These plasmids encode PRRSV-AGs containing T-cell epitopes, which are conserved through viral strains, i.e., RdRp, NSP1β and M [8]. The M sequence is included in a GP4GP5M chimera previously used in the context of DC- targeting [13] and the immuno-dominant B cell antigen N is also used to probe the antibody response. The PRRSV-AGd are either targeted to XCR1 via VB platforms using pig XCL1 as a targeting unit (pXCL1-GP4GP5M, pXCL1-N, pXCL1-NSP1β1, and pXCL1-NSP1β2) or untargeted (UT, pGP4GP5M, pN, pNSP1β, pRdRp). The PRRSV-AG sequences were derived from the PRRSV-1 13V091 strain (FL13, [12]). On day 0 (D0), plasmids encoding targeted or non-targeted PRRSV-AGs were administered either with EP or PJ, according to Table 1. As a difference with ref. [9], the DNA was not combined to cationic PLGA nanoparticles (see introduction). On D34, the attenuated FL13 MLV strain was administered by the intramuscular route. The IFNγ T-cell response to overlapping peptides covering the different vaccine PRRSV-AGs was assayed using the ELISPOT technique from PBMCs collected on D49 (i.e., D15 after the MLV-FL13b boost). The response against RdRp was evaluated with two peptide pools (N and C-terminal), in order not to exceed 50 peptides per pool and avoid toxicity. The MLV-FL13b and DNA+MLV-FL13b groups showed statistically significant IFNγ T-cell responses above the unimmunized control group to the N, GP4GP5M and RdRp2 peptide pools (Figure 3A–E). Few animals per group presented responses to the NSP1β peptide pools as a difference to in our previous study where the response was higher [9]; the use of cationic PLGA NPs (see the discussion) or difference in genetic make-up of the pigs between studies could explain this discrepancy. Higher responses to N and RdRp2 peptide pools were obtained in the PJ group as compared to in the MLV-FL13b group, with no clear benefit of DC-targeting (Figure 3A–E). The number of peptide pools recognized per pig was calculated (Figure 3F). Whereas five out of nine pig PBMCs responded to at least two peptide pools in the MLV-FL13b group, this proportion appears higher in the DNA+MLV-FL13b groups, such as in the XCL1-PJ group (eight out of nine pigs, Figure 3F). However, the increase over the MLV-FL13 group was not statistically significant for any of the DNA-MLV-FL13b group. In conclusion of this experiment using naked DNA administered with surface EP or PJ, the T cell response was slightly improved, but not significantly, by the DNA-MLV prime-boost strategy over the MLV-FL13b alone.

### 3.3. Antibody Responses Induced by the Prime-Boost DNA-MLV-FL13b Strategy

The anti-N IgG response in the sera was measured using a commercial ELISA kit [20]. The anti-N IgG response was already detected at the time of MLV injection, showing that a single DNA administration with EP and PJ is efficient at inducing anti-N responses. As previously found in ref. [9], the anti-N IgG response was clearly enhanced in the DNA+MLV-FL13b groups as compared to in the MLV-FL13b-only (Figure 4, at least *p* < 0.05 in all comparisons and timing). A higher proportion of pigs presented S/P ratios over 10 in the XCL1 groups than in the UT groups indicating that the XCR1-targeting of N is beneficial to the anti-N IgG response (Figure 4, *p* < 0.01 at D58). Sera collected at D58 were assayed for NAb detection using a classical in vitro neutralization assay with MARC-145 cells and FL13b virus, as well as with an experimental ELISA with PRRSV envelope proteins (ELISA-Env, PRRSV-1). None of the sera demonstrated neutralizing activity in our assay with MARC-145 cells. However, as shown Figure 5, all DNA-MLV groups presented higher titers in the ELISA-Env assay than the MLV-only group (*p* < 0.05). The discrepancy between the in vitro assay with MARC-145 and the ELISA-Env assay may be explained by the higher sensitivity of the ELISA test. Notably the ELISA titers were heterogeneous between pigs, possibly due to genetic differences in the antibody repertoires between pigs. This finding indicates that the DNA priming favors the generation of antibodies directed to antigenic determinants of the PRRSV-1 MLV envelope.

### 3.4. Protection Induced by the Prime-Boost DNA-MLV-FL13b Strategy against a Heterologous Challenge with FL07

So far, the increase of the immune responses induced by the DNA-MLV-FL13b strategy over the MLV-FL13b-induced responses reflects immuno-potentiating properties, which might be beneficial to cross-protection against a heterologous PRRSV-1 strain. FL13 shows a high genetic divergence to all, currently publicly available complete genomes of PRRSV-1 viruses and appears isolated in the phylogenetic tree, in a distinct branch from the Flanders’ 07V063 strain (FL07) [12]. The mean amino-acid conservation between the variable GP2, GP3, GP4, GP5 proteins of FL07 and FL13 is 84.9%, which is a usual variation used to assess heterologous protection [21]. Pigs were challenged intra-nasally with FL07 on D63. The body temperature was monitored with subcutaneous sensors. A slight rise in temperature (°C) from 38.3 ± 0.3 to 39.4 ± 0.4 (mean ± sem) was measured on D65 in the control non-vaccinated group and this rise was delayed to D66 in the MLV group, to 39.4 ± 0.6 (Appendix A). The rise was lower in the DNA-MLV-FL13b groups, and especially in the PJ groups (*p* < 0.05, Appendix A). The pigs did not present any pathological symptoms upon daily monitoring. FL07 RNA was measured on D58, 69, 71, 73, 76, 83 in the sera and in the nasal secretions using a specific qRT-PCR assay which was checked to be unable to detect FL13 (Appendix A and Figure 6). The area under the curve (AUC) of viral FL07 RNA detection (TCID_50_eq/mL) over time was calculated for each pig. MLV-FL13b significantly reduced the viral RNA content in serum (542 ± 168) and nasal secretion (74 ± 124) compared to in the serum of non-vaccinated pigs (12810 ± 4624 and 1254 ± 1796, for serum and nasal secretion respectively). However, the DNA-MLV-FL13b regimen did not enhance the protective effect of MLV-FL13b (Appendix A and Figure 6). The Ab levels detected with the ELISA-Env were enhanced by FL07 infection at D88 versus D58 before infection (compare Figure 5 and Appendix A), indicating that FL07 infection further boosted the Ab response against PRRSV-1 envelope determinants, without differences between vaccinated groups. However effective NAbs against FL13b in classical in vitro neutralization assay were not detected at D71 or D88, indicating that this FL13 DNA-MLV prime-boost strategy, followed by heterologous challenge FL07 was not efficient at inducing neutralizing or cross-neutralizing Abs.

### 3.5. Lack of Immune Correlates of Protection Induced by the MLV and Prime-Boost DNA-MLV-FL13b Strategies against the Heterologous FL07 Challenge

The MLV and DNA+MLV-vaccinated pigs demonstrate a variability in their B and T-cell immune responses and in the viral shedding values which could be useful to identify correlates of protection against a heterologous challenge. The different immunological, body temperature at D65 and viral variables were loaded on a principal component analysis (PCA) in order to identify the main axes of the data’s variance as well as correlated variables. The Dim 1 axis, which explains 31.32% of the variance, clearly separates the data between the control and the vaccinated groups (Figure 7A). The PCA loading of each variable shown in Figure 7B reveals that the total IFNγ T-cell response contributes most significantly to the Dim 1 axis, whereas the RNA viral data contribute most to the Dim 2 and little to the Dim 1 axis. None of the measured immunological data clearly opposes the viral data, indicating that IFNγ T-cell, anti-N and anti-env Ab responses are not correlates of heterologous viral protection, which was also confirmed by a Spearman analysis (2 by 2 analyses). The IFNγ T-cell response data appear to oppose the hyperthermia data at D65, which contributes moderately to Dim 1, and these variables significantly anti-correlate, although to a low degree (Spearman analysis, *p* = 0.01, *r* = −0.35). The PCA loading of the experimental data of the vaccinated groups only (excluding the controls) provided the same conclusions. Overall the statistics analysis of the experimental data shows that the level of heterologous protection, achieved in the absence of detectable NAbs to the MLV vaccine, is not correlated to the measured immune parameters, i.e., IFNγ T-cells, anti-N and anti-env responses. 

## 4. Discussion 

In this work, we showed that a DNA-MLV-FL13b prime-boost strategy using naked DNA administered with surface EP and PJ has a slight positive effect on the IFNγ T-cell response breadth and enhances the anti-N and envelope IgG response magnitude over the ones induced by MLV- FL13b-only, suggesting that the prime-boost strategy has an immuno-potentiation effect, especially on the B-cell arm of the response. However, this vaccine strategy did not improve the viral protection against a heterologous distant PRRSV strain over the one induced by MLV-FL13b alone. 

In a parallel work [9], the DNA prime promoted more clearly the breadth of the T-cell response induced by MLV-FL13b than in this present work. In that parallel study, the DNA was combined to cationic PLGA NPs, which also promoted local inflammation (unpublished, [14]). We propose that NPs combined to DNA has a positive adjuvant effect over naked DNA and that this adjuvant effect was needed to optimize the T-cell response triggered by the DNA-MLV prime-boost strategy. However, the precise mechanisms of the adjuvant effect of NPs, which might reside in an improved sensing of the DNA, remain to be investigated. 

Interestingly, naked DNA promoted the B-cell response induced by MLV-FL13b, whether the DNA was administered with surface EP or with PJ. Importantly, PJ administration was more efficient than EP to achieve in vivo transfection in skin, was less inflammatory, and was as potent as EP to elicit anti-N IgG responses (see Figure 4, D34). Needle-free jet delivery thus appears as a promising method to deliver DNA vaccines in pigs. Interestingly, the anti-Env Ab response was also enhanced by the prime-boost strategy, indicating that the Ab response to envelope determinants was promoted by the DNA prime. Such priming could have been induced by the GP4GP5M chimera, which includes a highly variable linear neutralizing epitope [22], or by the different PRRSV constructs through T helper activation. We favor this latter hypothesis as we did not detect a benefit of DC-targeting with the GP4GP5M chimera on anti-Env IgG responses, whereas it was obtained with the N antigen. However, the benefit of DC-targeting on the anti-N IgG response is less potent in the present study than in the parallel study [9] where NPs were combined to DNA. Again, NPs may have had an adjuvant effect, which has been shown to be necessary to obtain a positive effect of DC-targeting on the antibody and especially on the T-cell responses in many instances [23,24]. NPs combination to DNA in PJ delivery would need to be evaluated as the limited injection volume of the device may lead to precipitation of DNA/NP complexes and the high-pressure delivery may lead to deformation of NPs. Other adjuvants, less compelling than NPs, should be considered for DNA administration with PJ, either genetic or pharmacologic adjuvants, such as plasmids encoding CD40 ligand or GM- CSF or resiquimod [25] or by expression of RIG-I agonists produced during the DNA vaccine transcription process [26]. This positive effect of the DNA-MLV prime-boost on the antibody response could be used for the generation of protective antibody responses, ideally for cross-strain protection. However, epitopes involved in the generation of broadly neutralizing antibodies have not yet been identified, but evidence supports their existence [27]. 

Few works have evaluated the advantages of a DNA prime-boost to ameliorate PRRSV vaccination. A DNA-inactivated PRRSV or recombinant adenovirus prime-boost improved the anti-GP5 response in mice [28,29] but this approach has not been translated to pigs. The DNA-MLV prime-boost with a plasmid encoding N or truncated N in pigs was shown to correct the negative immunomodulatory properties of a PRRSV-2 MLV, by reducing the IL-10 and Treg responses generated by MLV administration [30]. This strategy nevertheless did not improve the viral protection against a heterologous challenge with a highly pathogenic strain [30]. We were not able to detect IL-10 production in our re-stimulation assays with overlapping peptides, possibly because a PRRSV-1 MLV, such as MLV-FL13b, was not able to induce an immunomodulatory IL-10 response as a difference with PRRSV-2 MLVs [31].

The priming with our DNA vaccines coding PRRSV-AG appeared to slightly reduce the increase in body temperature detected at D2 post-heterologous challenge (D65) with FL07, especially in the PJ groups. However, the rise in temperature induced by FL07 was very mild (1 °C) and clinical symptoms were not detected in our Large White pigs. In the initial description of FL07 by one of us, the pathology of FL07 was also limited [12]. In addition, the TCID_50_eq copy numbers/mL in blood was lower than the ones in other studies with similar strains [32,33]. The relatively low viral load in FL07 may explain the quite substantial, yet incomplete, cross-protection induced with MLV-FL13b alone which reduced by at least five-fold the overall viremia. The efficacy of PRRSV MLV vaccines also depends greatly on the degree of genetic similarity between the vaccine and challenge strains. A genomic analysis of the ORF2-6 reveals that the amino acid homology between the MLV-FL13b and the challenge FL07 strains was 84.9%, and 88% for GP5, what illustrates that the two strains were heterologous. 

The immuno-potentiation achieved by the DNA-MLV prime-boost was not sufficient to improve the incomplete cross-protection induced by MLV-only. Among hypotheses, it can be proposed that CD8^+^ T-cell responses were not sufficiently elicited by our strategy and that CD8^+^ T-cells are more instrumental to protection than CD4^+^ T-cells [34]. Indeed, the IFNγ T-cell ELISPOT assays that we performed do not differentiate whether the responding cells are CD4^+^ and CD8^+^ T-cells, and cytotoxicity may be a more important function than IFNγ synthesis. In addition, we may not have used the best T-cell antigen(s) to achieve high cross-strain protection. NSP5, which appears as a major immuno-dominant and well conserved T-cell antigen across PRRSV strains, would have been an interesting candidate [35]. However, because we wished to evaluate the targeting of T-cell antigens to DCs, we excluded NSP5 which contains many transmembrane domains and is thus not compatible with expression in VBs. Finally, an optimal cross-protection may require the contribution of broadly NAbs [27], which we could not detect to be induced by our strategy (see below). 

We were not able to identify the immune parameters associated with the partial heterologous protection induced by MLV-FL13b. Either we are missing pertinent parameters such as the CD8^+^ T-cell response, or protection could be controlled by the combination of immune parameters, possibly independently of their magnitude. It remains possible that the timing of the immune parameter measurement, especially the T-cell response, may have affected the correlation study. Indeed, we measured the IFNγ T-cell response 15 days after the boost, in order to compare the results with our previous work [9] and also because this timing was identified as an optimal time-point in another DNA-MLV prime-boost study [36]. We also measured the T-cell response after the challenge, but the spontaneous IFNγ responses were very high and impeded any interpretation. Finally, we found that a partial heterologous protection was achieved in the absence of detectable NAbs at the challenge time-point. PRRSV has often been reported to induce very low NAb titers which appear late in infection [37,38] or which were even non-detectable upon infection or MLV administration in several cases [39,40,41,42,43,44]. The non-detection could be due to the currently used in vitro techniques which may not capture the neutralization of the virus occurring in vivo. We used the classical MARC-145 cell-based NAb assay, which involves a simian kidney cell line that does not express the porcine CD169 and CD163 PRRSV receptors. Therefore, it is possible that the differences in receptor engagement between the in vitro assay and the in vivo situation is associated with the lack of NAb detection, as proposed in the case of flaviviruses [45]. Finally, porcine macrophage-based assays are difficult to parametrize, and cannot be used in the case of MLV-13b which does not grow in these cells. Research efforts are urgently needed to further develop system vaccinology tools, in particular, functional and refined biochemical antibody assays, to improve our understanding of the parameters involved in PRRSV control in order to develop the next generation of PRRSV vaccines as well as vaccines against other pig diseases.

Our study reveals that a DNA-MLV prime boost strategy holds promises for the control of PRRSV in pigs and it opens the way for further developments. The delivery with PJ allows a good transfection but minimal inflammation suggesting that it should be combined with complementary adjuvants. The effect of other T-cell antigens, such as the conserved NSP5 T-cell antigen, should also be evaluated. Finally, although much further down the road, broadly neutralizing epitopes, which appear to exist but remain to be molecularly identified [27], could be expressed in DNA vaccines associated with an MLV boost including shuffled structural genes [46].

## Figures and Tables

**Figure 1 viruses-11-00576-f001:**
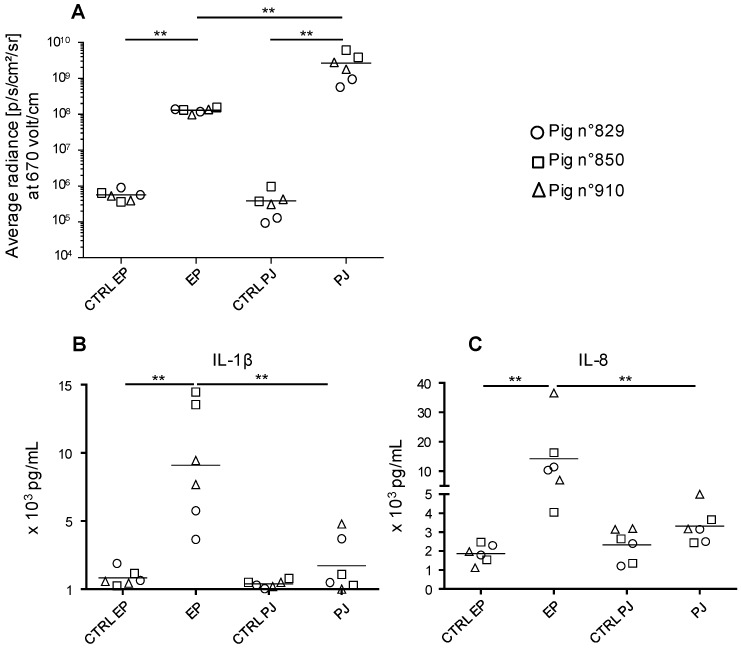
Expression of luciferase activity and cytokine secretion upon in vivo transfection of pig skin cells with pLuc delivered using surface electroporation (EP) and Pharmajet Tropis-ID device (PJ). Three Large White pigs (3-month-old, number 829, 850, and 910) were anesthetized. In the inguinal zone, 100 µg pLuc was injected intradermally in 100 µL saline and the skin site was subjected to surface EP as described in the Material and Methods. In the thoracic zone, 100 µg pLuc was administered with PJ in 100 µL saline. After 24 h, 8-mm biopsies were harvested at the sites of administration. (**A**) Biopsies from the transfected sites (independent duplicates) were processed for luciferase activity detection. Control skin of the same respective sites (inguinal and lower thoracic) were collected and treated in parallel and are designated as CTRL EP and CTRL PJ respectively. The luciferase activity results (p/s/cm^2^/sr) of the transfected and control skin from pig 829, 850, and 910 are indicated by a distinct symbol (circle, square, and triangle respectively) and means are shown. (**B**,**C**) Biopsies from the transfected sites (independent duplicates) were placed in culture medium for 24 h. Secreted IL-1β (**B**) and IL-8 (**C**) from each biopsy (same symbols as in **A**) were measured with a cytometric bead assay multiplex assay and the mean is shown. Statistically significant differences between 2 groups were calculated using the Mann–Whitney non-parametric test, ** = *p* < 0.01.

**Figure 2 viruses-11-00576-f002:**
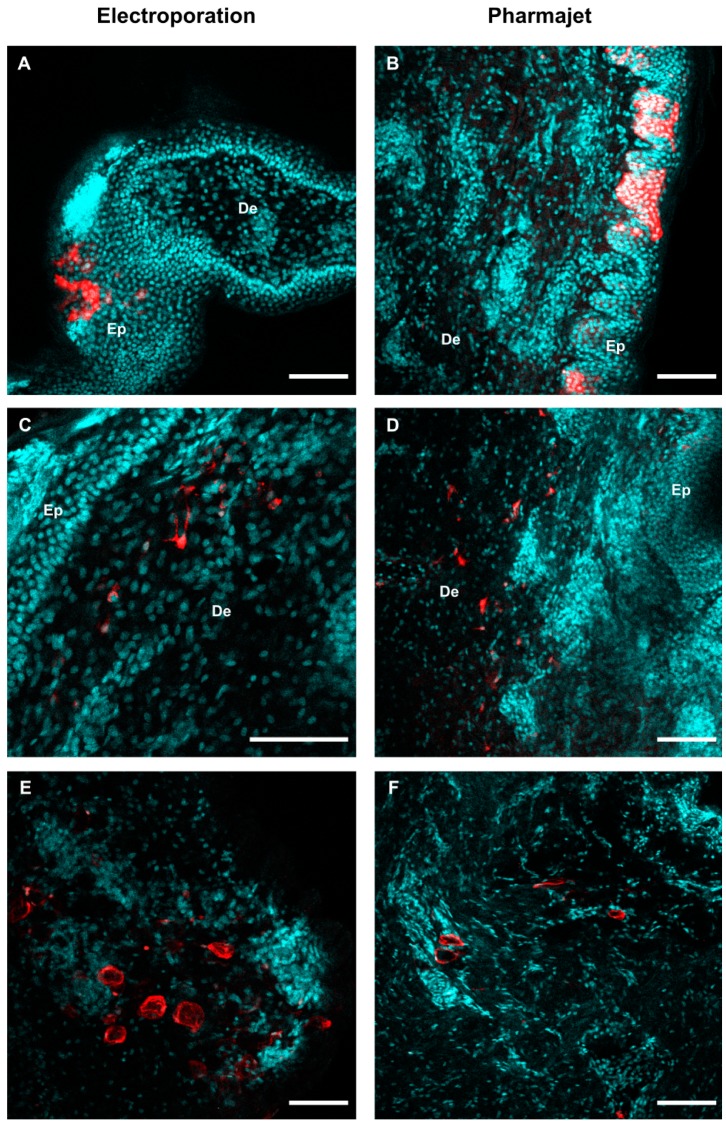
Transfection of skin cells with EP and PJ. Skin biopsies were harvested 24-h post- administration of 100 µg pmCherry with EP or PJ as described in Figure 1, and immuno-stained in toto using anti-mCherry rabbit IgG and Alexa Fluor 594-conjugated anti-rabbit IgG for mCherry detection (red) and DAPI for nucleus visualization (cyan) before tissue clearing. Representative optical sections were extracted from 2-photon acquisitions performed through thickness of the whole sample. (**A**,**C**,**E**) Expression of mCherry upon EP delivery, in the epidermis (**A**) and dermis (**C** where cells resemble fibroblasts and **E** where cells resemble adipocytes). (**B**,**D**,**F**) Expression of mCherry upon PJ delivery, in the epidermis (**B**) and dermis (**D** where cells resemble fibroblasts and **F** where cells resemble adipocytes). De: Dermis, Ep: Epidermis. Scale bar: 100 µm.

**Figure 3 viruses-11-00576-f003:**
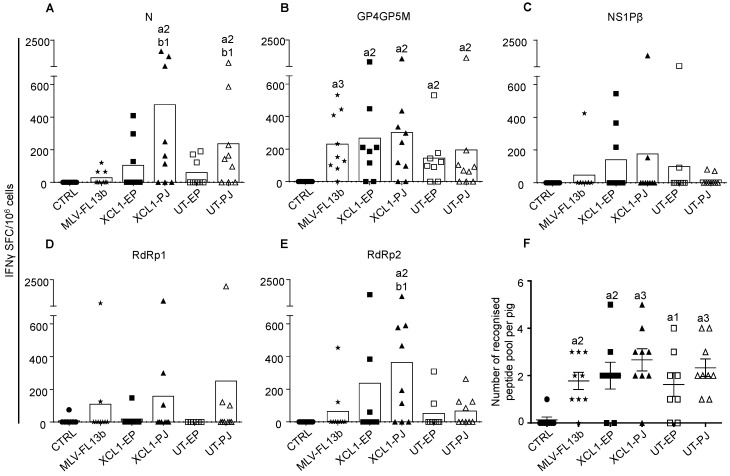
IFNγ T-cell responses induced by MLV-FL13b and DNA+MLV-FL13b. PBMCs were collected on D49 and they were analyzed for IFN**γ** T-cell response upon re-stimulation with overlapping peptide pools (10 µg/mL) in ELISPOT assays. (**A**–**E**) The responses to the N, GP4GP5M, NSP1β, RdRp1 (peptide 1 to 40) and RdRp2 (peptide 41 to 80) peptide pools are shown. An irrelevant peptide (10 µg/mL) was used as control. The mean number of spots from triplicates of stimulated minus control wells are shown (see Material and Methods). The mean number of spots across pigs per group is shown as a box. The plasmids used to immunize each group are listed in Table 1. (**F**) The number of recognized peptide pools per pig is reported (mean ± sem). Statistically-significant differences between 2 groups were calculated using the Mann–Whitney non-parametric test. The letter a indicates significant differences with the non-vaccinated control group, and b with the MLV-only group. Number 1 corresponds to *p* < 0.05, 2 to *p* < 0.01, 3 to *p* < 0.001.

**Figure 4 viruses-11-00576-f004:**
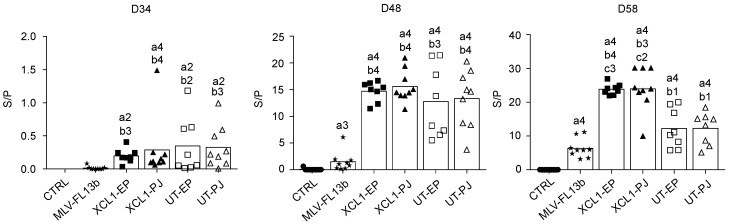
Anti-N IgG responses induced by MLV-FL13b and DNA+MLV-FL13b. Sera were collected on the indicated days and analyzed at a 1:40 dilution for anti-N IgG detection with the Ingezim PRRS 2.0 kit. The S/P ratios are reported and mean (box) is shown per group. Statistically significant differences between 2 groups were calculated using the Mann–Whitney non-parametric test. The letter a indicates significant differences with the non-vaccinated group, b with the MLV-only group, c with the respective UT group. Number 1 corresponds to *p* < 0.05, 2 to *p* < 0.01, 3 to *p* < 0.001, 4 to 0.0001.

**Figure 5 viruses-11-00576-f005:**
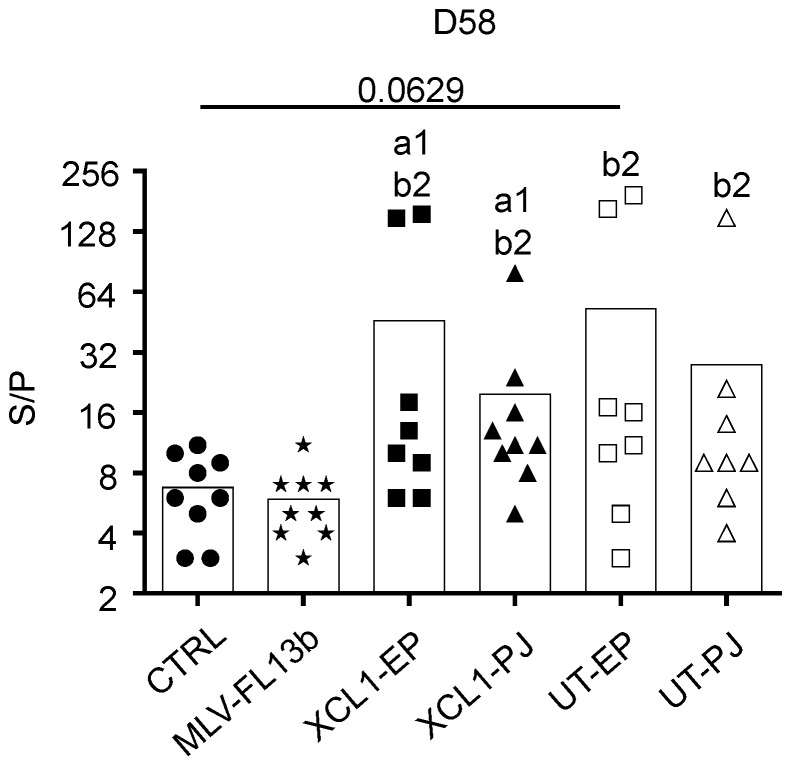
Anti-Env IgG responses induced by MLV-FL13b and DNA+MLV-FL13b. Sera collected at D58 were analyzed at a 1:20 dilution with the ELISA-Env. Statistically-significant differences between 2 groups were calculated using the Mann–Whitney non-parametric test. The letter a indicates significant differences with the non-vaccinated group, b with the MLV-only group. Number 1 corresponds to *p* < 0.05, 2 to *p* < 0.01. The *p*-value is indicated when close to significance.

**Figure 6 viruses-11-00576-f006:**
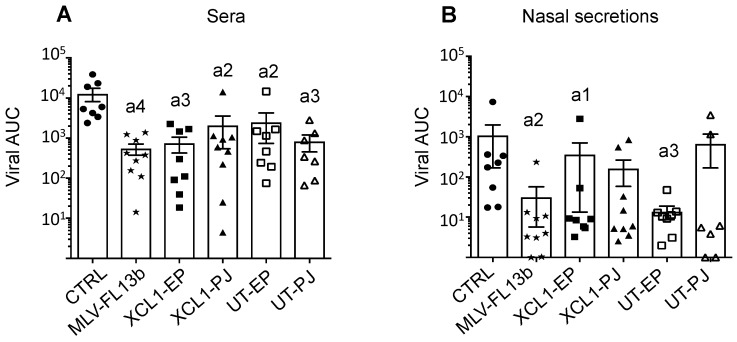
Protection induced by MLV-FL13b and DNA+MLV-FL13b against a FL07 heterologous challenge. The vaccinated pigs were subjected to FL07 challenge on D63 and the FL07 virus was selectively detected in sera (**A**) and nasal swab fluids (**B**) with specific qRT-PCR on D58 (before challenge), 69, 71, 73, 76, and 83. TCID50-equivalent/mL (TCID50eq/mL) was calculated with a viral standard curve (see Material and Methods). The limit of detection for FL07 was calculated as being 0.2 TCID_50_eq/mL. The area under the curve (AUC) for each pig was calculated and the mean (box) ± sem is shown per group. The letter a indicates significant differences with the non-vaccinated group (non-parametric–Mann-Whitney). Number 1 corresponds to *p* < 0.05, 2 to *p* < 0.01, 3 to *p* < 0.001).

**Figure 7 viruses-11-00576-f007:**
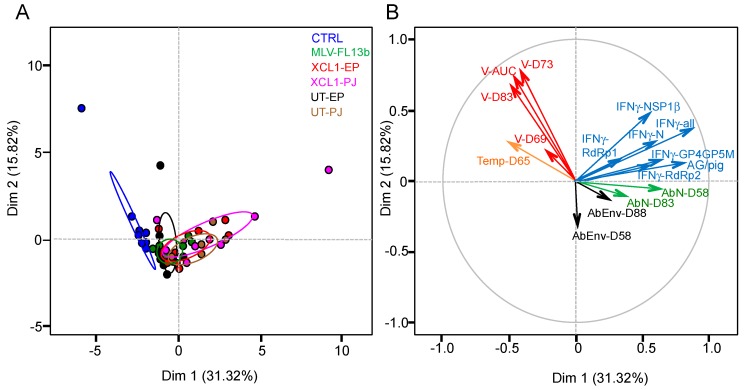
Principal component analysis (PCA) of the experimental variables of the immunization and the heterologous challenge. (**A**) PCA plot of the responses of pigs is depicted, with each pig represented as a dot in a specific colour according to its group assignment (see legend). (**B**) the PCA loading of each individual variable (16 factors) is shown and includes the individual viral RNA TCID_50_eq/mL at D69 (V-D69), D73, and D83, the viral RNA AUC (V-AUC), the individual body temperatures at D65 (Temp-D65), the IFNγ T-cell responses at D49 to the N (IFNγ-N), GP4GP5M, NSP1β, RdRp1, and RdRp2 pools, the total number of spots per pig (IFNγ-all), the number of T-cell antigens recognized per pig (AG/pig), the anti-N IgG at D58 (AbN-D58), and D83 and the anti-envelope Ab at D58 (AbEnv-D58) and D88. Dim 1 is the axis representing the highest percent of the variance and Dim 2 represents the second one.

**Table 1 viruses-11-00576-t001:** Vaccination with DNA vectors, MLV-FL13b and infection with the FL07 heterologous PRRSV strain.

Vaccines and Dates	Groups					
	Ctrl ^1^	MLV-FL13b	DNA+MLV-FL13b			
			XCL1-EP	XCL1-PJ	UT-EP	UT-PJ
DNA D0			pXCL1-N ^2^ pXCL1-GP4GP5M pXCL1-NSP1β1 pXCL1-NSP1β2 pRdRp	pXCL1-N pXCL1-GP4GP5M pXCL1-NSP1β1 pXCL1-NSP1β2 pRdRp	pN pGP4GP5M pNSP1β pRdRp	pN pGP4GP5M pNSP1β pRdRp
MLV-FL13b D34	- ^3^	+ ^4^	+	+	+	+
FL07 challenge D63	+	+	+	+	+	+

^1^ non-vaccinated group, ^2^ plasmids per pig inoculated at D0, ^3^ not injected, ^4^ injected.

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
