# Peer review of "A DNA Prime Immuno-Potentiates a Modified Live Vaccine against the Porcine Reproductive and Respiratory Syndrome Virus but Does Not Improve Heterologous Protection"

_viruses, 2019, doi:10.3390/v11060576_

Round 1
Reviewer 1 Report
The manuscript is well written and structured. Methodologies are appropriate. The Authors should investigate the reason of the lack of heterologous protection. Alternatively, they should give a more detailed explanation of the results.
Reviewer 2 Report
The work described in the manuscript is of great interest. A few minor details are lacking; The authors should provide the list of the peptides used for the stimulation in ELISPOT, perhaps as a supplementary material. The absence of neutralizing antibody in the study pigs , all of which were boosted with the MLV, is unusual. This observation deserves better explanation. In addition, theoretically, it would be interesting if a group that was primed and boosted with DNA vaccine had been included. This would have have shown if the DNA induced immune reponse, which must mostly be CMI response, is sufficient for any amount of protection against viremia and nasal shedding.
Reviewer 3 Report
Bernelin-Cottet et al. report the testing of a vaccination regimen against porcine reproductive and respiratory syndrome virus (PRRSV) in pigs. With the aim to induce cross-reactive immunity, pigs were primed with DNA encoding conserved T-cell epitopes of PRRSV and 34 days later vaccinated with a modified live vaccine (attenuated PRRSV-1, subtype 1, strain 13V091). A challenge was performed 29 days later with the FL07 strain of PRRSV-1, subtype 1. For DNA priming, the authors compared surface electroporation and needle-free jet injection and evaluated the benefit of XCR1-targeting. Read-outs following vaccination and challenge were 1) clinical signs, 2) viral loads in sera and nasal swab fluids, 3) T-cell response upon re-stimulation (IFN-γ ELISpot), and 4) PRRSV-specific antibody (ELISA).
The study – as reported in this manuscript – is an important contribution to vaccine research against PRRSV infection in pigs. It shows that DNA (coding for conserved T-cell epitopes) priming increased mainly B-cell responses and to a lesser extent T-cell responses but did not increase protection in terms of viral load upon challenge. Also, it describes needle-free jet delivery as a method to deliver DNA vaccines in pigs.
The reviewer strongly supports publication of this data, but is uncertain if publication as a separate manuscript is warranted. This comment has also been made to the editors.
Main comment:
Could the authors please state why this data should be published separately from the submitted companion manuscript (submitted manuscript 1) “Assessing the immunogenic properties of DNA vaccines delivered with nanoparticles, electroporation and dissolvable microneedle patches in pig skin”
Some minor comments are listed in the following:
a) Line 52+53: sentence repeated
b) Lines 179 and 180: please state the size of the biopsies and please give specifications for the “antibiotic/antimycotic cocktail”
c) Line 207: please specify “antiprotease cocktail”
d) Line 237: replace “were” by “was”
e) Line 275: replace “envelop” by “envelope”-> check spelling throughout manuscript
f) Line 316: replace “were” by “was”
g) Line 336: replace “on” by “in”
h) Line 370: replace “pool” by “pools”
